Biochemical characterization and inhibition of thermolabile hemolysin from Vibrio parahaemolyticus by phenolic compounds

Vazquez-Morado Luis E. 1 2
Robles-Zepeda Ramon E. 1
Ochoa-Leyva Adrian 2
Arvizu-Flores Aldo A. 1
Garibay-Escobar Adriana 1
Castillo-Yañez Francisco 1
Lopez-zavala Alonso A. alexis.lopez@unison.mx 1
1 Departamento de Ciencias Quimico Biologicas, Universidad de Sonora , Hermosillo , Sonora , Mexico
2 Departamento de Microbiología Molecular, Instituto de Biotecnología, Universidad Nacional Autónoma de México , Cuernavaca , Morelos , Mexico
Uversky Vladimir
Electronic publication date: 2021 Jan 6
Publication date: 2021
Volume: 9
Electronic Location ID: e10506
Received 2020 Sep 21; Accepted 2020 Nov 16
Copyright: ©2021 Vazquez-Morado et al.
Copyright year: 2021
Copyright holder: Vazquez-Morado et al.
License: This is an open access article distributed under the terms of the Creative Commons Attribution License, which permits unrestricted use, distribution, reproduction and adaptation in any medium and for any purpose provided that it is properly attributed. For attribution, the original author(s), title, publication source (PeerJ) and either DOI or URL of the article must be cited.
License URL: https://creativecommons.org/licenses/by/4.0/

Keywords: Thermolabile-hemolysin, SGHN phospholipases, Phenolic compounds, Inhibition, Molecular docking, Thermal stability

Funding: The Universidad de Sonora grant USO313004740 The UNAM-CIC-UNISON-2018 and -2019 grants for the academic exchange program National Council on Science and Technology (CONACYT) M.Sc. scholarship This research was funded by the Universidad de Sonora grant: USO313004740, the UNAM-CIC-UNISON-2018 and -2019 grants for the academic exchange program. Luis E. Vazquez-Morado received a National Council on Science and Technology (CONACYT) M.Sc. scholarship. The funders had no role in study design, data collection and analysis, decision to publish, or preparation of the manuscript.

==============================
Vibrio parahaemolyticus (Vp), a typical microorganism inhabiting marine ecosystems, uses pathogenic virulence molecules such as hemolysins to cause bacterial infections of both human and marine animals. The thermolabile hemolysin VpTLH lyses human erythrocytes by a phospholipase B/A2 enzymatic activity in egg-yolk lecithin. However, few studies have been characterized the biochemical properties and the use of VpTLH as a molecular target for natural compounds as an alternative to control Vp infection. Here, we evaluated the biochemical and inhibition parameters of the recombinant VpTLH using enzymatic and hemolytic assays and determined the molecular interactions by in silico docking analysis. The highest enzymatic activity was at pH 8 and 50 °C, and it was inactivated by 20 min at 60 °C with Tm = 50.9 °C. Additionally, the flavonoids quercetin, epigallocatechin gallate, and morin inhibited the VpTLH activity with IC50 values of 4.5 µM, 6.3 µM, and 9.9 µM, respectively; while phenolics acids were not effective inhibitors for this enzyme. Boltzmann and Arrhenius equation analysis indicate that VpTLH is a thermolabile enzyme. The inhibition of both enzymatic and hemolytic activities by flavonoids agrees with molecular docking, suggesting that flavonoids could interact with the active site’s amino acids. Future research is necessary to evaluate the antibacterial activity of flavonoids against Vp in vivo.

Introduction

Vibrio parahaemolyticus (Vp) is a Gram-negative bacterium naturally found in marine ecosystems, inhabiting high-valuable species such as fish and shrimps. Recently, Vp has been implicated in high mortalities in shrimp culture ponds, causing significant worldwide economic losses (Cuéllar-Anjel, 2012). Several disease control strategies were assayed in shrimp culture, such as an increased water exchange, phage therapies, probiotics, and supplemented food with prebiotics and antibiotics (Dy, Rigano & Fineran, 2018; Freire-Moran et al., 2011). The primary treatment for Vp’s infections is based on antibiotics use; however, uncontrolled administration interferes with beneficial host-microbiota promoting multidrug bacterial resistance (Santos & Ramos, 2018; Zeng et al., 2019). Thus, it is necessary to develop alternative treatments against Vp by identifying novel potential molecular targets (Li et al., 2019; Perez-Acosta et al., 2018). Massive sequencing technologies paired with biochemical studies have identified Vp virulence elements such as adhesion factors, type III and VI secretion systems, proteases, hemolysins, and others (Li et al., 2019).

Vp hemolysins are extracellular toxins that lyse the erythrocytes using several molecular mechanisms. Most Vp strains express several hemolysins, such as thermostable direct hemolysin (TDH) and TDH-related hemolysin (TRH). These hemolysins have been characterized using structural, biochemical, clinical, and epidemiological approaches (Raghunath, 2014; Saito et al., 2015). Both TDH and TRH are pore-forming toxins, while thermolabile hemolysin (TLH) has phospholipase activity (Shinoda et al., 1991). The tlh gene encodes the TLH, a full-protein of 418 amino acids (MW ≈ 47.3 kDa), and a post-transductional modification removes the N-terminal signal peptide, leaving a mature protein of 399 amino acids (Taniguchi et al., 1986). The TLH from Vp “VpTLH” was initially described as a hemolytic factor activated by lecithin, with phospholipase A2 (PLA2)/lysophospholipase enzymatic activity (Shinoda et al., 1991; Yanagase et al., 1970). PLA2 hydrolyzes glycerophospholipids at the sn-2 position to release lysophospholipids, which has a powerful detergent capacity and participates in cell signaling that can induce apoptosis (Flores-Diaz et al., 2016; Murakami & Kudo, 2002). TLH is widely distributed among Vibrio species, such as Vibrio anguillarum, Vibrio vulnificus, Vibrio alginolyticus, Vibrio harveyi, and others (Klein et al., 2014; Wang et al., 2007). TLH research is mainly focused on using this gene as a biomarker to identify virulent strains in epidemiology studies. However, structural and biochemical reports of TLH in comparison with TDH and TRH are scarce. Thermolabile hemolysins from V. harveyi and V. alginolyticus (VhTLH and VaTLH, respectively) were virulence factors against fish (Jia, Woo & Zhang, 2010; Zhong et al., 2006). V. vulnificus TLH (VvTLH) lost both enzymatic and hemolytic activity after 30 minutes at 55 ∘C and 65 °C, respectively. This enzyme also showed cytotoxic activity against flounder gills cells and in vivo toxicity with a medium lethal dose of 1.2 µg of protein per gram of flounder (Zhong et al., 2006). Recombinant Va TLH was also toxic when injected into zebrafish with a lethal dose (0.8 µg/ gr). Li, Mou & Nelson (2013) reported that Vibrio alguillarum secretes a TLH with potent hemolytic specific activity against rainbow trout erythrocytes (Li, Mou & Nelson, 2013).

TLH has a high-conserved amino acid sequence (>90%) among Vibrio species and contains a characteristic GDSL motif belonging to the esterase-lipases family (Akoh et al., 2004). This motif is located in the C-terminal domain (residues 151-406) of the VpTLH sequence (Taniguchi et al., 1986). TLHs belong to the serine-proteases family and contain a catalytic triad composed of serine 153, aspartic acid 154, and histidine 393 (numbering is according to the VpTLH sequence), which is located in the SGNH hydrolases domain (Carter & Wells, 1988; Taniguchi et al., 1986). The SGNH domain comprises four conserved blocks, I, II, III, and V, based on many esterases/hydrolases enzyme‘s catalytic mechanism. Briefly, block-I comprises the typical GXSXG motif found in lipases/esterases, in which Ser153 acts as the nucleophile during catalysis. While glycine 204 and asparagine 248 in blocks II and III are the proton donors in the oxyanion cavity. On the other hand, the histidine 393 located in block V activates the catalytic residue Ser153, and the aspartic acid 390 stabilizes the tetrahedral intermediate, ensuring the correct orientation during catalysis (Akoh et al., 2004; Upton & Buckley, 1995; Wan, Liu & Ma, 2019).

Plants mainly synthesized the phenolic compounds as secondary metabolites, such as phenolic acids (caffeic acid, gallic acid, among others) and polyphenols (p.e. flavonoids: quercetin, rutin, morin, among others) (Cheynier, 2012; Panche, Diwan & Chandra, 2016). These compounds function as antioxidants, cytotoxic, antifungal, antibacterial, and enzyme inhibitors destabilizing the cell membrane (Özçelik, Kartal & Orhan, 2011; Petrescu, Paunescu & Ilia, 2019).

The quercetin inhibits the enzymatic activity of PLA2 of the snake Crotalus durissus terrificus venom via hydrogen bonds and hydrophobic interactions with the enzyme active site (Cotrim et al., 2011). Furthermore, morin and rutin were potent PLAs inhibitors of Crotalus atrox and Crotalus durissus cascavella, respectively; but rutin also inhibits PLA2 from the porcine pancreas (Iglesias et al., 2005; Lindahl & Tagesson, 1997). Other phenolics compounds as gallic, ferulic, caffeic acids, and epigallocatechin gallate inhibited both enzymatic activity and cytotoxic activity Crotalus durissus cumanensis PLA2 (Pereanez et al., 2011). VpTLH displays PLA2 enzymatic activity and has similar active site amino acids (serine, histidine, and glutamic/aspartic acid) found in several venom snake PLA2 (Murakami & Kudo, 2002). Zhao et al. (2020), reported that resveratrol (stilbene group of polyphenols) efficiently inhibits VhTLH hemolytic activity and cytotoxicity directly by binding to the active site. However, a high resveratrol dose shows tissue accumulation in fish (Takifugu rubripes), resulting in toxic effects (Zhao et al., 2020). Therefore, additional research will provide information about the potential use of phenolic compounds derivatives to inhibit pathogenic factors as TLH.

As previously mentioned, most studies have focused on TDH and TRH and its role in human infections by V. parahaemolyticus. However, only a few studies have characterized the biochemical properties of VpTLH, and there are not reported enzyme inhibitors. Contrary, phenolic compounds are effective inhibitors against several PLA2 and VhTLH. Therefore, these compounds could also inhibit VpTLH activity providing novel alternatives for treating Vp infections. In this work, the effect of pH and temperature on the enzymatic activity of recombinant VpTLH was evaluated, and the kinetics parameters were determined using p-nitrophenyl laurate (PNPL) as substrate. Additionally, we also analyzed several phenolics compounds as inhibitors of both enzymatic and hemolytic activity, and we describe the possible molecular interactions using in silico molecular docking.

Materials & Methods

All reagents were ACS, electrophoresis, or molecular biology grade as required and were purchased in Merck (Sigma-Aldrich). Exceptions were indicated in the text.

Cloning the VpTLH gene and recombinant protein expression

The nucleotide sequence of VpTLH used in this study was obtained from the gene bank accession number AB012596.1. VpTLH was obtained as a synthetic gene (Atom®) and cloned into pET-28b (+) plasmid, adding the C-terminal 6x-His tag for the purification process. Chemically competent E. coli BL-21 strain rosetta II cells were transformed with VpTLH plasmid by thermal shock and incubated in SOC media (tryptone 2% w/v, yeast extract 0.5% w/v, 10 mM NaCl, 2.5 mM KCl, 10 mM MgCl2, 10 mM MgSO4 and 20 mM glucose) at 37 °C by 4 h. Bacterial cells were plated in Luria-Bertani agar plates supplemented with kanamycin (25 µg/ml) at 37 °C overnight for the plasmid selection. After that, a single colony was inoculated in 5 ml of the antibiotic-LB medium by four h at room temperature; this culture was scaled up to 50 ml under the same conditions and incubated overnight. Subsequently, a Fernbach flask containing 1 l of LB medium added with kanamycin (25 µg/ml) was inoculated with the 50 ml culture and incubated at 37 °C and 220 rpm. When the optical density reached ≈ 0.6 units (λ = 600 nm), we added IPTG (Isopropyl β-D-1-thiogalactopyranoside) to a final concentration of 1 mM, inducing the overexpression of VpTLH. The Fernbach flask was maintained in an orbital shaker (200 rpm) for 16 hours at 25 °C. Bacterial cells were recovered by centrifugation at 7,000 rpm for 20 minutes at 4 °C, and the pellet was washed using 0.7% NaCl and spun as before. The supernatant was discarded, and the bacterial cell pellet was stored at −80 °C until use.

Protein purification and in vitro refolding

VpTLH was recovered from frozen pellet, which was resuspended in lysis buffer (50 mM Tris base, 1 mM DTT, 5 mM Benzamidine, 5 mM EDTA, 100 mM NaCl; pH 7.0) at ratio 1:8 (w/v). Bacterial cells were lysed by sonication on an ice bed with six pulses of 5-s and 5-s rest at 30 % amplitude. The homogenate was clarified at 12,000 rpm for 20 min at 4 °C, and SDS-PAGE (12%) stained with blue-coomassie was used to analyze the protein expression in soluble and insoluble fractions (Laemmli, 1970). Several overexpression conditions were analyzed, but in all cases, VpTLH was obtained as inclusion bodies. Therefore, inclusion bodies were isolated from insoluble cellular debris; which was resuspended by sonication (as before) using buffer 1 (50 mM Tris base, DTT 1 mM, 5 mM EDTA, 2% Triton x-100; pH 7.0) at ratio 1:4 (w/v), and the homogenate was centrifugated at 12,000 rpm for 20 min. The precipitate was recovered, and centrifugation was repeated three times. After that, the pellet was 2-times washed in buffer 2 (buffer 1, without Triton X-100) under the same conditions. Finally, the recovered inclusion bodies were solubilized using urea (50 mM Tris base, DTT 1 mM, urea 8M, pH 7.0) by sonication and incubated overnight at 4 °C with constant stirring. The homogenate was clarified (12;000 rpm at 4 °C for 30 min), and the soluble fraction containing the VpTLH in urea 8M was recovered. The protein concentration was quantified at 280 nm in a nanodrop®equipment (ϵ ≈ 96,510 M−1 cm-1). VpTLH was purified by Immobilized Nickel Affinity Chromatography on an Äkta Prime (GE) under denaturing conditions at 25 °C. Therefore all buffers used during this process contained 8 M urea. Briefly, 1 ml HiTrap®IMAC-HP column was equilibrated with five volumes of buffer A (50 mM Tris base, 500 mM NaCl, 8 M urea; pH 7.4), then 4 ml (15 mg denatured protein) of VpTLH solution was loaded into the column. Non-bounded proteins were washed with buffer A until absorbance (λ = 280 nm) reached the base-line. Bound proteins were eluted with a linear gradient of buffer B (buffer A + 500 mM imidazole), eluted fractions were collected, and SDS-PAGE analyzed. Fractions containing a single band with a molecular weight of 48.3 kDa were pooled and refolded by dialysis with a cut-off membrane of 12–14 kDa at 4 °C. Refolding was carrying out by sequentially decreasing urea concentration (4, 1 M and without urea) in buffer 50 mM Tris-Base buffer at pH 7.5 (250 µl by 6 hours each). Buffer without a chaotropic agent was changed twice (12 hours each), then the protein solution was removed from membrane dialysis, centrifuged, and stored at 4 °C.

Enzymatic and hemolytic activity assay of the VpTLH

Enzymatic activity was measured spectrophotometrically using the lipase/esterase assay described by Nawani, Dosanjh & Kaur (1998), which was modified to measure the activity of TLH in the presence of egg yolk lecithin as an enzyme activator (Shinoda et al., 1991). Each reaction was conducted in a final volume of 1 ml containing: 50 mM Tris-HCl pH 7.5, 0.0001% egg yolk lecithin, 200 µM p-nitrophenyl laurate (PNPL), and the reaction was started by addition of 10 µL of refolded VpTLH (6.2 µM final concentration). PNPL hydrolysis was measured at a wavelength of 410 nm (PNPLε410nm = 11.8 cm−1 M−1) for 5 min at a temperature of 37 °C in a Cary 50®UV-VIS spectrophotometer (Varian). Negative control assay consisted of the same reaction without VpTLH. Specific activity was calculated using the equation: (1) U∕mgprotein=m⋅V∕ε⋅p⋅l

where m is the slope of the reaction; V, the reaction volume; p the protein concentration (mg/ml) and l, cell path-length in cm.

Hemolytic activity of VpTLH was quantified using human erythrocytes as a substrate (Malagoli, 2007). Erythrocytes were extracted from human blood, which was kindly donated by one male volunteer, who was previously informed about the extraction procedure, the use, and the proper disposal of blood samples, according to the established protocol by the Institutional program of environmental health and safety of the Universidad de Sonora. Also, the volunteer signed an informed consent about her/his participation in this research. Briefly, erythrocytes were isolated from Human blood by centrifugation at 700 rpm for 5 minutes at 4 °C. The plasma was discarded, and erythrocytes were washed three times (0.9% saline solution); they were then carefully resuspended in saline phosphate buffer (PBS; 8 mM Na2HPO4, 2 mM KH2PO4, 37 mM NaCl, 2.7 mM KCl, pH 7.4) to 1:200 ratio. Hemolytic activity was measured by the release of hemoglobin at 540 nm and expressed as a percentage of hemolysis using Tween 20 (0.2%) as a positive control (100% hemolysis). Erythrocytes autolysis (without enzyme or detergent) was also recorded and subtracted in each assay. Each assay was conducted with 300 µL of the erythrocyte suspension, 0.0001% egg yolk lecithin, and VpTLH (0.23 mM final concentration), PBS was added to a final volume of 1 ml. Reaction tubes were carefully homogenized and incubated at 37  °C for 60 min after tubes were centrifuged (2 min at 700 rpm), and the supernatant was recovered, and absorbance was recorded at wavelength of 540 nm. All measurements were carried out by triplicate.

Biochemical parameters of VpTLH

To determine the pH effect on VpTLH enzymatic activity, we used several buffer solutions varying pH values (5.5, 6, 7, 7.5, 8, 8.5, 9, 10, and 11) under the standard assay mentioned before. PNPL molar extinction coefficient (ε) was used for each of pH values: 1.97, 6.52, 11.8, 12.75, 13.9, 14.3, 14.6 and 14.6 (cm µ−1 M−1), respectively (Kademi et al., 2000). Sodium citrate (pH 6), Tris-HCl for pH 7 to 9, and sodium carbonates (pH 10 and 11). All buffer’s concentration was kept constant at 100 mM. The temperature on VpTLH activity was assayed in standard conditions at pH 8.0 by varying temperature assay from 10–80 °C, increasing by 10 °C. Results were expressed as a percentage of residual activity. The temperature with the highest Vp TLH activity was used as 100% of residual activity. Furthermore, VpTLH activation energy (Ea) was calculated by plotting linearized Arrhenius equation: (2) lnk=lnA−Ea∕R∕T

where slope m = −Ea/R; k, initial velocities; T, temperature (kelvin) and R, universal gas constant (J/molK). A negative control without enzyme in each pH and temperature condition was assayed to discard substrate precipitation or chemical hydrolysis,

For the thermostability test, the enzyme was incubated 15 min at temperatures from 10 to 80 °C with 10 °C intervals. Then enzymatic activity was measured under standard conditions. Residual activity was calculated as a percentage at VpTLH showed the highest activity (100%). Additionally, VpTLH melting-temperature (Tm) was obtained by Boltzmann sigmoidal analysis using Prism 5 software (GraphPad®).

Determination of the VpTLH Michaelis-Menten parameters

The kinetics parameters km and Vmax of VpTLH were determined from the initial velocities by varying PNPL concentrations from 20 to 400 µM using the enzymatic standard conditions mentioned before. Initial velocities were recorded for 2 minutes and were adjusted to the Michaelis-Menten non-linear regression model using the Prism 5 program (GraphPad®). All measurements were carried out by triplicate. Additionally, the Michaelis-Menten constant (Km), Vmax, and turnover number (kcat) of the enzyme were calculated (Michaelis et al., 2011).

Enzymatic and hemolytic activity inhibition assays

The following phenolic acids were used: gallic acid (GA), vanillic acid (VA), protocatechuic acid (PR), and chlorogenic (CL). The following flavonoids were used: quercetin, morin, rutin, and epigallocatechin gallate (EGCG) added to the standard enzymatic assay. All flavonoids were used to a final concentration range: 1–20 µM, while phenolics acids were evaluated at 30 µM and 100 µM. Meanwhile, quercetin, morin, and EGCG were evaluated as an inhibitor for VpTLH hemolytic activity, which was added to the standard hemolytic assay to a final concentration range of 1–20 µM. VpTLH residual activity (enzymatic or hemolytic) in the presence of phenolic compounds was calculated as a percentage of VpTLH activity in the absence of phenolic compounds. Flavonoids showed the highest inhibitory effect (see results section), were selected to calculate inhibitor concentration required to reduce 50% of enzyme activity (IC50) using a concentration range of 0.1–50 µM. The data were normalized using VpTLH in absence of inhibitor as 100% of enzymatic activity; then analyzed with the dose-response variable slope model based in Hill Slope equation as described in Prism-5 software (GraphPad®).

Comparison of the TLH sequences from different Vibrio species and homology modeling of the VpTLH

The amino acid sequence of the VpTLH, with access code Q99289 in the UniProt database, was compared with that of other Vibrio species, including Vibrio cholerae (Q9KMV0), Vibrio alginolyticus (C7EWQ8), Vibrio harveyi (Q2XPT2), Vibrio anguillarum (A0A191WA34) and Vibrio vulnificus (A0A1V8MSL8). These were compared using the ClustalW algorithm (File S1). Based on the alignment obtained, the conserved regions of the sequences were identified, and the presence of the GDSL and SGNH motif, which are specific domains of this type of enzymes. The VpTLH homology model was built using free access algorithms such as PHYRE2, I-TASSER, SWISS-MODEL, and commercial software MOE (File S2). As we expected, all the predicted structures based on 6JL1 crystal structure (TLH from V. vulnificus), showed that the overall structure of predicted models is quite similar in RMSD = 0.324 Å. PHYRE2 model (intensive mode) showed the lowest RMSD (0.223 Å). Therefore this model was used for structural analysis and molecular docking simulations. Structural analysis was performed in UCFS Chimera 1.13.1 (Pettersen et al., 2004) and CCP4-MG programs (McNicholas et al., 2011).

Molecular Docking of the VpTLH substrates and Inhibitors

Both natural (phosphatidylcholine, PC), synthetic (PNPL) substrate and best IC50 inhibitors were docked into Vp TLH active site using AutoDock Vina algorithm in UCFS Chimera 1.13.1 program (McNicholas et al., 2011; Trott & Olson, 2010). Before docking experiments, both ligands and protein were analyzed in DockPrep function to minimize structure, partial charges calculation, and hydrogen atoms were also added. The 3D structure files of the PNPL, quercetin, morin, and EGCG compounds were obtained from the PubChem database with access codes 74778, 5280343, 5281670, and 65064, respectively. In contrast, PC structural data were obtained from the MOE database. The fpocket2 algorithm predicted a putative ligand pocket cavity in the Phyre2 investigator program (Kelley et al., 2015; Le Guilloux, Schmidtke & Tuffery, 2009) and the Site Finder function of the MOE program. Sequence alignments located VpTLH active site amino acids and spatial coordinates were established by superposition with Vv TLH crystal structure (RMSD = 0.400 Å). VpTLH final docking area was established in coordinates X = 24.4143 Å, Y =  − 2.51601 Å and Z =  − 35.3974 Å (volume = 28,802.47 Å3). For each ligand, 20 poses were generated using Iterated Local search method supported in AutodDocvina (Trott & Olson, 2010); models with low free energy binding force function (ΔG) were analyzed in Discovery Studio 2019 program (Biova®).

Results

Purification and refolding of VpTLH

All the over-expression conditions produced the VpTLH (≈ 47 kDa) protein in the insoluble fraction (Fig. 1A, lanes 1–5). It was in agreement with no phospholipase activity detected on soluble protein fraction by an agar-plates assay using egg yolk lecithin as substrate (0.1 % egg yolk lecithin, 3% agar). The higher production of VpTLH was using 0.4 mM IPTG for 16 hours at 25 °C. After that, the inclusion bodies were solubilized in 8 M urea and purified by a single chromatographic step (IMAC) under denaturing conditions. As a result, we obtained a single peak elution at 115 mM imidazole (Fig. 1B), corresponding to a unique SDS-page band of VpTLH molecular weight (Fig. 1A, lanes 6 and 7). The VpTLH was refolded by dialysis, and the esterase activity using PNPL as a substrate on the soluble fraction confirmed that protein was active (Fig. 1C) with a specific activity of 0.47 U/mg of protein (Shinoda et al., 1991). Figure 1D showed that VpTLH enzymatic activity gradually increased, with the highest PNPL hydrolysis at 10 µg lecithin, suggesting that VpTLH could hydrolyze PNPL in lecithin, as early noticed. Furthermore, the enzymatic activity decreased by <50% in the presence of >50 µg lecithin. A quantitative hemolytic activity assay showed that 2–10 µg of purified VpTLH elicits 76–85 % relative activity (100% hemolysis was tween-20) (File S3). In this sense, we assayed both enzymatic and hemolytic activity using 10 μg of lecithin and confirmed that VpTLH is a lecithin-dependent protein.

Figure 1 Recombinant overexpression, purification, and refolding of VpTLH.

(A) 12% SDS-PAGE of the recombinant over-expression process and in vitro refolding of purified VpTLH. M, molecular weight marker; lane 1 and 2, the soluble and insoluble fractions of non-induced E. coli culture. Lane 3 and 4, insoluble and soluble fraction 16 hours after the addition of IPTG to the culture; lane 5, solubilized inclusion bodies (8M urea); lane 6, purified VpTLH under denaturing conditions; lane 7, in vitro refolded VpTLH. (B) Chromatogram of VpTLH IMAC purification under denaturing conditions. (C) Esterase activity assay of refolded Vp TLH. 410 nm absorbance increases as PNPL hydrolysis releasing p-nitrophenol. The assay was performed by triplicate; PNPL self-hydrolysis (control) was assayed without Vp TLH. (D) Effect of lecithin on VpTLH enzymatic activity.

pH and temperature effect on VpTLH activity

VpTLH maximum activity was detected at pH 8.0 (100%), slightly decreasing at pH 8.5 (97% residual activity). Enzymatic activity suddenly decreased at pH <7.5 and >9.0. Contrary, the VpTLH was inactive at acidic medium (pH 6.0) while showed low activity (<20%) at alkaline pH (Fig. 2A). Amino acid sequence analysis indicated that VpTLH was thermolabile hemolysin (Nishibuchi et al., 1989). Interestingly, to our knowledge, there is no report evaluating the effect of temperature on VpTLH activity. First, we found that activity increased from 10 °C (10% residual activity) to a maximum level at 50 °C. At higher temperatures (>50 °C), the VpTLH enzymatic activity rapidly decreased, and at 80  °C, no esterase activity was found (Fig. 2B). These data were analyzed using the linearized Arrhenius equation by plotting natural logarithm (initial velocities) against the inverse of each temperature in Kelvin degrees (Fig. 2C). This analysis showed a single inflection point (50 °C) with linear activity decreasing as a temperature increase (until 80  °C), suggesting that temperature above 50 ∘C drastically affects VpTLH enzymatic activity. The Arrhenius equation calculates the activation energy (Ea), which is the energy that VpTLH requires to hydrolyze PNPL, resulting in Ea = 26, 688 kJ/mol. Additionally, we evaluated the temperature stability by incubating the enzyme from 10 to 80 °C for 20 min. After that, the enzyme retained >80% residual activity at temperatures below 40 °C. At 60 °C the enzyme only retained 6 % of residual activity, being inactive at 70 and 80 °C. The data were adjusted to the Boltzmann sigmoid model (R2 = 0.9985), calculating VpTLH Tm = 50.94 °C (Fig. 2D).

Figure 2 Biochemical properties of Vp TLH.

Enzymatic activity was calculated as the residual activity respect to the highest value detected in each assay. Results were the mean ± SE (n = 3). (A) pH effect on enzymatic activity, a different buffer, was a function of pH evaluated as described in the ‘Materials and Methods’ section. (B) Activity profile at different temperatures. Cell holder temperature within the reaction cell was stabilized by 60-sec min each assay. (C) The plot of linearized Arrhenius equation, a temperature in which enzymatic activity starts decreasing (inflection point), was fitted to a linear model (R2 = 0.985). lnK, the natural logarithm of initial velocities; temperatures were in Kelvin degrees. (D) Thermal stability of VpTLH, data were fitted to the Boltzmann sigmoidal model (R2 = 0.99). All data were analyzed in Prism5 Graphpad® program.

VpTLH Michalelis-Menten parameters

Initial velocities were measured using PNPL as a substrate from 20 to 400 µM (Fig. 3) and showed a typical Michaelis-Menten profile by plotting initial velocities vs. [PNLF]. Data were adjusted to non-linear regression analysis with a correlation factor R2 = 0.9851. After that, we obtained the VpTLH kinetics parameters, a Vmax = 0.7736 U/mg (± 0.041) and a km = 0.151 mM (±0.017) (Fig. 3). Also, the enzyme turnover number (kcat) was 37.37 s-1 (±1.97 SD) (Table 1). TLHs Michalelis-Menten kinetic parameters reports are scarce; recently Vv TLH kinetics constants were determined using a fluorogenic substrate (Red/Green BODIPY PC-A2) (Wan, Liu & Ma, 2019) and other authors reports PLA2 activity (from snake venom) using 4N3OBA as substrate (Pereanez et al., 2011). In spite of differences in substrates chemical composition used in each report, we observe that VpTLH has the highest Vmax and turn over compared with other TLH and other PLA2 enzymes, lower substrate affinity (k m) than V. vulinificus TLH (Wan, Liu & Ma, 2019). Table 1, shows that substrate affinity have variable magnitude among compared enzymes, which is closely-related to substrate differences used (Eisenthal, Danson & Hough, 2007). Meanwhile, Wicka et al. (2016), reported similar substrate affinity and kcat for cold-adapted GDSL-lipase from Pseudomonas sp. S9 using p-nitrophenyl butyrate which has short carbon chain in fatty acids substituent than PNPL (Wicka et al., 2016).

Figure 3 Effect of substrate concentration (PNPL) on Vp TLH enzymatic activity.

Fitting data calculated Michaelis-Menten kinetics parameters to non-linear regression model (R2 = 0.9851). All substrate concentrations were assayed by triplicate.

Table 1 Michaelis-Menten kinetic parameters of TLH and other enzymes with similar catalytic properties, including GDLS-lipases and snake venom PLA2.

Enzyme and source	km (mM)	Vmax(U/mg)	kcat(s−1)	Reference	
TLH Vp	0.151	0.7736	37.37	This work	
TLH WT Vv	0.020	0.0216	0.051	Wan, Liu & Ma, 2019	
TLH G389D Vv	0.0209	0.0118	0.028	Wan, Liu & Ma, 2019	
sPLA2Cdc	60	0.0034	NR	Pereañez et al., 2009	
sPLA2Cdt	31	0.0082	NR	Oliveira et al., 2002	
EstS9N Psp.	0.161	NR	3.31	Wicka et al. (2016)	
Notes.

NR not reported

EstS9 Psp. GDSL-lipase Pseudomonas sp

Cdc C. durissus sub cascavella

Cdt Crotalus durissus terrificus

Polyphenols inhibited both VpTLH enzymatic and hemolytic activity

As shown in Fig. 4, phenolic acids GA, PR, and VA inhibited the VpTLH activity by 20% at 30 µM, while CL does not affect activity compared to control (assay without phenolics acids). Increasing phenolics compounds to 100 µM did not affect the VpTLH activity (p < 0.05) (Fig. 4). Also, rutin did not affect the activity at all evaluated concentrations; while, quercetin, morin, and EGCG at 20 µM (the highest concentration) decreased the activity by 70%, 65%, and 67%, respectively (Table 2). At low concentration (1 µM), morin was the most effective to inhibit VpTLH (30% inhibition), whereas, at 10 µM, both quercetin and EGCG were also able to reduce activity by 60%. These results suggest that flavonoids were more suitable to inhibit VpTLH phospholipase activity. Additionally, the dose-response analysis showed that quercetin was the best-evaluated inhibitor (IC50=4.51 µM). EGCG and morin also exhibited similar IC50 values: 6.290 µM and 9.914 µM, respectively (Fig. 5).

Figure 4 Effect of phenolic acids on Vp TLH enzymatic activity.

Vp TLH activity was assayed in the presence of each phenolic acid and the final concentration as indicated. Residual activity was calculated based on Vp TLH activity under optimal assay conditions in the absence of phenolics acids. Results are mean SE (n = 3) statistical differences (p < 0.05) compared to control without phenolics acids as denoted with an asterisk. Control (-), VpTLH without phenolic acids; GA, gallic acid; PR, protocatechuic acid; CL, chlorogenic acid and VA, vanillic acid.

The phospholipids are abundant in erythrocytes’ membrane cells; thus, VpTLH hemolytic activity in flavonoids at the same concentrations used in enzymatic inhibition experiments (1–20 µM) was assayed. However, the effect of phenolic compounds on VpTLH hemolytic activity was not evaluated. All flavonoids gradually diminished hemolysis, increasing its concentration, compared to control without flavonoids (Fig. 6). Low concentration (1 and 5 µM) did not significantly affect erythrocytes lysis, but quercetin and EGCG at 10 and 20 µM inhibited the VpTLH hemolytic activity 15% and 30%, respectively. Morin achieved only 15 % inhibition at the highest evaluated concentration. We could not evaluate flavonoids at concentration >20 µM because precipitation of hemolytic assays components was observed.

Table 2 Inhibition percentage of Vp TLH enzymatic activity by flavonoids.

Compound/Dose
(µM)	1	5	10	20	p valuea	
Quercetin	14.9 ± 4.2	34.5 ± 4.0	62.2 ± 7.4	70.3 ± 8.5	0.011*	
Morin	30.3 ± 2.1	38.7 ± 8.3	45.4 ± 6.2	65.5 ± 5.4	0.018*	
EGCGb	16.8 ± 7.2	44.7 ± 6.8	60.3 ± 13.0	67.8 ± 7.2	0.020*	
Rutin	NIc	NI	NI	NI	——	
Notes.

a Statistically significant values (p < 0.05) are represented with an *.

b Epigallocatechingallate.

c NI, not inhibit.

Figure 5 Dose-response analysis of Vp TLH inhibition by flavonoids.

Vp TLH enzymatic activity was assayed (n = 3) in presence of each flavonoid; data were fitted (R2 > 0.95) to the dose-response model to calculate IC50 values. Residual activity was calculated as a percentage considering VpTLH enzymatic activity in the absence of tested flavonoids as 100%. EGCG, Epigallocatechingallate.

Figure 6 Inhibition of Vp TLH hemolytic activity by flavonoids.

Each inhibitor concentration was assayed in triplicate. Bars represented SEM. Hemolytic activity in the presence of flavonoids was calculated as percentage respect to VpTLH without inhibitors (CN). Inhibitor concentrations with statically significant differences (p < 0.05) compared to control are denoted with an asterisk.

Homology modeling and docking analysis indicates that VpTLH has conserved folding and an active site cavity suitable to bind both substrates and inhibitors

We aligned the VpTLH amino acid against TLHs of other pathogenic Vibrio species such as Vibrio alginolyticus (Va), Vibrio harveyi (Vh), Vibrio campbelli (Vc), Vibrio cholerae (Vch), Vibrio diabolicus (Vd), and Vibrio anguillarum (Van). After that, the VpTLH maintained a high-sequence identity >80 % with Vd, Va, Vh, and Vc; while Vv, Van, and Vch showed a lower sequence identity being 73%, 65%, and 64 %, respectively (File S1). This analysis showed that VpTLH has hydrolase/esterase superfamily well-conserved GDSL and SGNH motifs (Akoh et al., 2004; Upton & Buckley, 1995), as was previously reported in other TLHs (Jang et al., 2017; Jia, Woo & Zhang, 2010; Wang et al., 2007). Two main domains comprised the TLH sequence, the N-terminal domain included from amino acid residue 24 to 133 (signal peptide 1–23), and C-terminal (also called SGNH domain) comprised 134–418 amino acids (numbering were according to the VpTLH sequence). The N-terminal’s biological function is not well defined, while SGNH-domain is directly related to enzymatic function divided into four blocks that contain invariable catalytic residues (Akoh et al., 2004). GDSL motif is located in block I (139-158) and contained catalytic serine residue (Ser153), while that Gly204, Asn248, and His393 are found in blocks II, III, and V, respectively. SGNH hydrolases have conserved catalytic triad His-Ser-Asp. This last amino acid residue was also found in VpTLH block V (Asp390), and Gly’s substitutions were found in Va and Vv TLHs (Jang et al., 2017; Li, Mou & Nelson, 2013).

VpTLH homology model was built in PHYRE2 using Vv TLH crystal structure (PDB: 6JL1) as a template since its share >74% sequence identity with VpTLH, and the resulting model showed an excellent superposition with the template (RMSD = 0.256 Å) (Fig. 7A). N-terminal domain (109 amino acid residues) was composed of β-sheets and three small α-helices exposed to solvent while the sizeable C-terminal domain (274 a.a) adopts a typical SGNH α/ β/ α folding related to phospholipase function. Ser-Asp-His catalytic triad and other active site amino acids were located in this domain and distributed in four well-conserved blocks of the SGNH superfamily (Fig. 7A) (Akoh et al., 2004). β-sheet central core flanked by α-helices compose this domain; all four blocks converge to form ligand pocket cavity as was predicted by the fpocket2 algorithm in the Phyre2 investigator program (Kelley et al., 2015; Le Guilloux, Schmidtke & Tuffery, 2009) and Site Finder function of the MOE program. We superposed the VpTLH model to the Vv TLH structure, and the catalytic triad was located in the pocket, suggesting it as the VpTLH active site (Figs. 7A and 7B). Although N-terminal domains remain close to the active site, no catalytic function was previously reported for this domain. Nucleophile Ser153 interacted with both Asp390 and His393 that comprised the conserved catalytic triad of serine hydrolases family (Fig. 7C). However, structural and biophysical experimental techniques are necessary to get a more precise model; crystallization assays are in progress in our laboratory with both apo and holo Vp TLH.

Figure 7 Predicted Structure of the Vp TLH.

(A) Overall structure superposition of the VpTLH and the VvTLH. N- and C-terminal domains are showed by magenta/cyan (Vp) and gray/orange (Vv). The cylinders colored by atom type shows the catalytic triad (Ser-His-Asp). (B) Superposition of the catalytic amino acids VpTLH (carbon atoms in gray) and Vv (carbon atoms are purple). (C) Hydrogen bonds (continuous lines) interactions in catalytic amino acids of VpTLH. (D) Charge surface representationVp TLH catalytic site cavity (indicated by the arrow).

Finally, we used molecular docking simulations of possible interactions with substrates and flavonoids into the C-terminal domain with area = 28.36 Å × 32.65 Å × 31.09 Å  following AutoDockVina protocols with centered predicted active site cavity (Fig. 7D). For each ligand, 20 interaction-models were constructed, and we selected the best solution based on the free-energy binding and the position inner the active site. The most favorable interacting-coupling score for substrates were: Phosphatidylcholine (PC) = −3.9 kcal/mol and PNPL= −4.5 kcal/mol. PC is a natural substrate for TLH and one of the most abundant phospholipids, including phosphatidylserine, in the cellular membrane.

Figure 8 shows the molecular docking and interaction diagram of PC and PNPL into Vp TLH active site. PC properly accommodates with the aliphatic chain (glycerol sn-1) buried inside the active site, while fatty acid (glycerol sn-2) was located in the active site surface and polar substituent (choline) is more exposed to the solvent. PC interacted with several active site amino acids by hydrogen bonds between Gln292 and OH-groups (both fatty acids) and Asn254 with phosphate group; also, Lys303 stabilize phosphate group by a saline bridge. Asn252, Ala206, and Tyr253 stabilize fatty acids chains and choline methyl groups by aliphatic C-H interactions. Non-canonical substrate interaction prediction showed an embedded PNPL into VpTLH active site. The p-nitrophenol ring shows hydrophobic π-alkyl- interaction to Ala206 (coordination), and the carboxylate was stabilized by hydrogen bonds with Tyr360 lateral chain OH-. 14-C fatty acid chain (laurate) was located in a similar arrangement as the second PC fatty acid substituent (sn-2). Interestingly, catalytic residues (SGNH) did not interact with both substrates in docking experiments under the used conditions. Similar results were also reported in the crystallographic model of Vv TLH single mutant (Gly389Asn) in complex with hexamethylene glycol, concluding that TLHs could elicit conformational flexibility upon substrate binding (Wan, Liu & Ma, 2019).

Figure 8 Molecular docking (A) and interaction maps (B) of substrates into Vp TLH active site.

PC, phosphatidylcholine and PNPL, p-nitrophenylaurate. The protein molecule is displayed as a surface in white and ligand as a cylinder colored by atom type with carbon atoms in green. Interaction maps were showed depicted by color as follows: hydrogen bonds (green), alkyl (pink), saline bridge (orange), and Van der Waals interactions (light green).

Flavonoids that inhibited both enzymatic and hemolytic activity (quercetin, morin, and EGCG) were also docked into the predicted VpTLH active site (Fig. 9A). Interactions diagram showed that free energy binding was favorable to EGCG > morin > quercetin with scores: −7.9, −7.2, and −6.4 kcal/mol, respectively. EGCG (gallic acid substituent) and quercetin (ring A) were buried into the active site as observed with both substrates. Ala206 π-alkyl stabilizes EGCG gallic acid and quercetin ring A, while hydrogen bridge Gln292 with quercetin 1′-oxygen and EGCG 4′-hydroxyl group (Fig. 9B). Such interactions were not observed in morin docking simulations, positioned near active site surface throw π-anion interaction with Glu300 (ring B and C) and hydrogen bridge to Thr297 and Van der Waals forces with oxygen groups located in morin ring B (Fig. 9B). Morin and quercetin have identical chemical formula and molar mass but differ in hydroxyl; morin is 2′, 4′ while quercetin is 3′, 4′ orientation in ring B; such could be related to differences in ΔG scores and active site interactions (Xiao et al., 2012). EGCG and quercetin displayed similar disposition into active site VpTLH, as observed with substrates evaluated. These results are consistent with the inhibition experiment; therefore, both flavonoids could be suitable compounds for chemical modification for structure/function studies and evaluate inhibition/attenuation capacity during V. parahaemolyticus infection.

Figure 9 Molecular docking (A) and interaction map (B) of flavonoids into the Vp TLH active site.

EGCG = epigallocatechin gallate. The protein molecule is displayed as a surface in white and ligand as a cylinder colored by atom type with carbon atoms in green. Interaction maps were depicted by color as follows: hydrogen bonds (green), π-alkyl (pink), π-anion (orange), unfavorable donor/acceptor hydrogen bond (red), and Van der Waals interaction (light green).

Discussion

We performed several strategies to obtain soluble VpTLH, such as testing several cultures and overexpression conditions and using a protein with and without a signal peptide. Unfortunately, in all strategies, the VpTLH was obtained as inclusion bodies. The VpTLH was refolded into active form eliminating chaotropic agents by dialysis, recovering 15 mg of purified active enzyme per liter of culture media. Contrary, several studies have been reported with different results. Shinoda et al. (1991) first reported the recombinant production of VpTLH as an active soluble protein from the periplasm of E. coli (Shinoda et al., 1991). Recombinant hemolysins TDH, TRH, and TLH from Vp also were expressed as inactive form and renatured by carbamide gradient dialysis (Zhao, Tang & Zhan, 2011). Despite these differences, recombinant VpTLH showed lecithin-dependent phospholipase and hemolytic activity as other Vibrio TLHs (Jia, Woo & Zhang, 2010; Li, Mou & Nelson, 2013; Zhao, Tang & Zhan, 2011). GDSL-esterases and SGNH-hydrolases enzymes have flexible active site exhibiting conformational changes upon substrate binding and favoring enzyme catalysis (Akoh et al., 2004; Wan, Liu & Ma, 2019). Lecithin could induce VpTLH local or global conformational changes in active site vicinity, allowing hydrolyzing PNPL, which will require further studies using biophysical and biochemical approaches to demonstrate this hypothesis.

Lecithin-dependent hemolysins are widely overexpressed between the Vibrionaceae family’s microorganisms, and they showed different temperature sensitivities. For example, V. anguillarum hemolysin has a broad optimal temperature from 37 to 64 °C (Li, Mou & Nelson, 2013), while the optimal temperature in V. harveyi hemolysin was 37 °C and it was inactivated by 30 min at 65 °C (Zhong et al., 2006). Miwatani et al. (1972) first describe that Vp strains secreted hemolytic factors showing different behavior with the temperature increase from 60 °C (partially inactivated) to 90 °C (entirely inactive) (Miwatani et al., 1972; Takeda, Hori & Miwatani, 1974). Later, Taniguchi et al. (1985) identified another hemolysin that was wholly inactivated by 10 min at 60 °C (Taniguchi et al., 1986).

Our results suggest that VpTLH maximum enzymatic activity was at 50 °C and suddenly decreases to entirely inactive at 80 °C, while gradually decreasing activity at low temperatures (10–40 °C) retaining 80% residual activity at 37 °C. The linearized Arrhenius equation (Fig. 2C) suggested that VpTLH follows a one-steady denaturation process without apparently intermediate transition states with Ea = 26.6 kcal/mol (Segel, 1975). This behavior also was described in psychrophilic enzymes; that show high structural flexibility to diminish activation energy during catalysis (Feller & Gerday, 1997). Furthermore, VpTLH lost 50% of enzymatic activity by 30 min at 50.9 °C and was inactivated at 70 °C. VpTLH melting temperature is related to linearized Arrhenius plot suggesting that temperatures >50 °C induce the loss of enzymatic activity by local (active site) or global structural destabilization. Finally, these results indicate that Vp TLH is a thermolabile enzyme; however, other thermodynamic and structural approaches are necessary to understand the TLH inactivation process.

TLH is a ubiquitous protein among Vibrionaceae species (Wang et al., 2007), and it is a molecular marker to both clinical and environmental V. parahaemolyticus strains (Bej et al., 1999; Chen et al., 2017). VpTLH can lysate both human and fish erythrocytes through phospholipase A2 activity and showed cytotoxicity activity against human cells (Wang et al., 2012; Wang et al., 2015). In this sense, we found that flavonoids were more effective than phenolic acid to inhibit VpTLH enzymatic activity and hemolytic capacity against human erythrocytes (Table 2 and Fig. 5). TLHs inhibition studies are scarce, but recently, resveratrol at 8 µg/ml inhibits almost 100% Vh TLH hemolytic activity by binding into the active site and at 2 µg/ml reduced cell damage caused by Vh TLH (Zhao et al., 2020). Resveratrol is a polyphenol belonging to stilbenes with a characteristic nucleus of 1,2-diphenylethylene that could have hydroxyl substitutions in aromatics rings, as occurring in flavonoids and other phenolics compounds, and therefore share biological and physicochemical activities with those (Han, Shen & Lou, 2007). Other studies are focused on antibody neutralization using phage display technologies (Wang et al., 2012). Polyphenols have inhibitory activity against PLA2 from snake venoms; Crotalus durissus terrificus PLA2 enzymatic activity was inhibited (40%) with 50 µM. Also, Iglesias et al. (2005) isolated PLA2 from tropical rattlesnake (C. durissus sub cascavella) and found that morin 20 µM reduce PLA2 activity by 70%. EGCG showed the best inhibitory activity against C. durissus sub. Cumanensis PLA2 compared to phenolics compounds as cafeic and ferulic acid. All evaluated compounds belong to the flavonol group (3-hydroxy flavone) that share a base structure with various biological properties depending on hydroxyl substitutions and different conjugations (Massi et al., 2017). These observations are consistent with our results because quercetin or quercetin-derivatives, such as glycosides and gallic acid or hydroxyl substitutions, inhibited the VpTLH. Quercetin (IC50 = 4.5 µM) was a 2-fold higher VpTLH inhibitor than morin (IC50 = 9.9 µM), which has a single change in one hydroxyl position in 3′ and 2′, respectively. While EGCG contains a 3′-gallic acid conjugation and an additional 5′-hydroxyl group, these substitutions could be associated with the slight decrease in inhibitory effect than quercetin, but it was more effective than morin. Quercetin 3-O-rhamnosylglucoside conjugation (rutin) may affect binding to VpTLH active site and thus could not inhibit the enzyme. Changes in hydroxyl group positions (3′, 4′ and 5′) could be affecting the VpTLH inhibitory capacity.

TLHs show high conserved amino acid sequence among Vibrios species (>70%) and maintains the characteristic catalytic triad (Ser-His-Asp) except for Vv and Van in which acidic residue was substituted by chloride atom during catalysis (Wan, Liu & Ma, 2019). We obtained a successful model of VpTLH using a recently solved Vv TLH crystal structure as a template (PDB: 6JL1). VpTLH has a C-terminal domain with typical GDSL α − β hydrolase folding, located at the active site (Akoh et al., 2004; Wan, Liu & Ma, 2019). Molecular docking experiments suggested that quercetin, EGCG, and morin could interact with VpTLH active site and PC and PNPL with free energy binding values from −3.9 and −7.9 kcal/mol. Zhao et al. (2020) reported similar results by docking resveratrol to Vh TLH and found that binding energy was - 6.0 kcal/mol. Resveratrol binds to Vh TLH active site through Lue247 and Tyr368 by π-alkyl interactions and hydrogen bonds, respectively (Zhao et al., 2020). Also, both residues are essential during the resveratrol binding process to Vh TLH and are straightly related to hemolytic activity inhibition. VpTLH interaction diagrams showed that Ala206 and Gln292 could be important for binding both substrate (PC) and inhibitors (quercetin and EGCG) to the active site by hydrogen bonds with Gln142 and π-alkyl interactions with Ala206. Thus, we suggest that both residues could be critical during the binding process to VpTLH. Although the study of such interactions could be the target of future studies using conjugated phenolics compounds that may enhance VpTLH inhibition, future biochemical and structure-function studies should examine these hypotheses.

In summary, our results showed that VpTLH has conserved GDSL hydrolase folding with conserved active site composed by catalytic triad Ser-His-Asp. Biochemical studies demonstrated that polyphenols as quercetin, EGCG, and morin were suitable VpTLH inhibitors, and molecular docking suggests the interaction with the active site. Future research should also focus on evaluating antibacterial or/and bacteriostatic effects of flavonoids on V. parahaemolyticus and bacterial infection in a host such as shrimp or fish.

Conclusions

In the present study, we purified in two steps from inclusion bodies a functional VpTLH. The enzyme showed thermolabile characteristics compared to TDH and TRH. Furthermore, the kinetic parameter km was similar to that described for other GDSL enzymes. On the other hand, quercetin, EGCG, and morin inhibited VpTLH activity possibly due to the active site binding, as predicted by molecular docking and showed similar structural orientation into active site Vp TLH compared to PC and PNPL. Therefore, the flavonoids that we evaluated and others with similar physicochemical properties could be suitable compounds to chemical modification for structure/function studies and to evaluate the inhibition/attenuation capacity during V. parahaemolyticus infection.

Supplemental Information

File S1 Sequences alignment of TLHs from genus Vibrio

All aligned sequences have the highly conserved GDSL domain (green), SGNH catalytic domain (light blue) and the blocks part of the phospholipase A2 domain (red).

Click here for additional data file.

File S2 Superposition of Vp TLH homology models with the Vv TLH experimental structure

Algorithms used to obtain structural models were as follows: MOE (blue), SWISS-MODEL (green), Phyre2 (pink) and I-TASSER (orange) and 6JL1 (white). General RMSD: 0.384. The RMSD of predicted models using as reference the experimental structure of Vv TLH were 0.145 for SWISS-MODEL, 0.253 for Phyre2, 0.407 for MOE, and 0.335 for I-TASSER.

Click here for additional data file.

File S3 VpTLH quantitative hemolytic activity

Tween-20 was considered as 100% hemolysis. Control (-), was buffer without in absence of TLH.

Click here for additional data file.

File S4 Biochemical and kinetic parameters raw data

Click here for additional data file.

File S5 Phenolics acid and flavonoids inhibition profile raw data

Click here for additional data file.

We thank Cesar Otero-León for technical support.

Additional Information and Declarations

Competing Interests

Author Contributions

Data Availability

The authors declare there are no competing interests.

Luis E. Vazquez-Morado conceived and designed the experiments, performed the experiments, analyzed the data, prepared figures and/or tables, authored or reviewed drafts of the paper, and approved the final draft.

Ramon E. Robles-Zepeda conceived and designed the experiments, analyzed the data, authored or reviewed drafts of the paper, and approved the final draft.

Adrian Ochoa-Leyva conceived and designed the experiments, performed the experiments, analyzed the data, authored or reviewed drafts of the paper, and approved the final draft.

Aldo A. Arvizu-Flores conceived and designed the experiments, performed the experiments, prepared figures and/or tables, authored or reviewed drafts of the paper, and approved the final draft.

Adriana Garibay-Escobar and Francisco Castillo-Yañez conceived and designed the experiments, authored or reviewed drafts of the paper, and approved the final draft.

Alonso A. Lopez-zavala conceived and designed the experiments, analyzed the data, prepared figures and/or tables, authored or reviewed drafts of the paper, and approved the final draft.

The following information was supplied regarding data availability:

The sequences are available in GenBank: AB012596.1.

The sequence alignment of TLH from genus Vibrio, the superposition of Homology model VpTLH using free access algorithms and commercial software and experimental structure, the VpTLH quantitative hemolytic activity, the biochemical and kinetics parameters (Figs. 1–3) raw data, and inhibition by phenolics acids and flavonoids (Figs. 4–6) are available in the Supplemental Files.

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
