# Peer review of "Biochemical characterization and inhibition of thermolabile hemolysin from Vibrio parahaemolyticus by phenolic compounds"

_PeerJ, doi:10.7717/peerj.10506_

## Round 0.1 · original submission · Minor Revisions

Please address critical points indicated by all reviewers and revise your manuscript accordingly

Reviewer 1 ·

Basic reporting

The study by Vazquez-Morado et al evaluated the biochemical and inhibition parameters of the thermolabile recombinant hemolysin (VpTLH) Vibrio parahaemolyticus

--The study highlights an important observation, but the organization of the manuscript is poor. Overall, the manuscript is poorly written, many typos at multiple places and inconsistent use of tenses.

Experimental design

Experimental design and methodology is good, well described, and with appropriate controls.

Validity of the findings

The conclusions drawn from the study are good and relevant to the prospective implications in the field.

Additional comments

Authors should work upon improving the language of the paper.

·

Basic reporting

For the most part, the article are written clearly. The biggest issue I have is with the rendering of the symbols (like micro, beta, etc). A lot are rendered simply as boxes. Please have this corrected, and be sure that no rendering error exist in the final version.

Here are the lines where I catch these symbol rendering errors:
Line 94, 95, 129, 437, 495, 546, 547, 554, 562

Other than the rendering error, there are a number of minor changes to the text that I would suggest:
- Line 65: not controlled to uncontrolled
- Line 84: delete "that" in "and also that participate in cell"
- Line 141: please correct the grammar for "and determined the kinetics Michaelis-
142 Menten parameters using p-nitrophenyl laurate (PNPL) as substrate"
- Line 154: gen to gene
- Line 158: platted to plated
- Line 181: precipitated to precipitate or precipitant
- Line 188: Niquel to Nickel
- Line 192: "to a constant non-bounded protein". I don't understand what this means. Please clarify
- Line 197: sequential to sequentially
- Line 199: please rephrase: "by overnight each" to make the sentence clearer
- Line 203: delete "by the" in "... measured by the spectrophotometrically..."
- Line 221: established to "established protocol"?
- Line 336: >50 mg lecithin. Is this supposed to be 50 microgram?
- Line 540: please correct "specie"
- Line 552: anti-body to antibody
- Line 557: caffeic to caffein
- Line 570: excepting to except for

Experimental design

The experiments are well designed, with numerous biophysical techniques used to purify the protein, find the rate of its enzymatic and hemolytic activities, and understand the interaction of the VpTLH with flavonoids through molecular modeling. The authors do a great job putting all the experiments together into a coherent study.

Validity of the findings

I have a few questions that I have regarding the data that hopefully will help improve the next interation of the paper:
- Line 279-280: "The data were analyzed with the dose-response model in Prism-5 software". Are there parameters that need to be inputted for the model to be built? Can the author provide more detail?
- Line 337-338: "A quantitative hemolytic activity assay showed that 2-10 ug of purified VpTLH elicits 76-85 % relative activity (100% hemolysis was tween-20)." Can this data be presented in the paper?
- For all figures with residual activity, is it possible for the magnitude of the rate at 100% activity be written next to the figures?
- Line 373: "VpTLH has the highest Vmax and turn over compared with other TLH". The Vmax, km and kcat is around an order of magnitude greater than the other TLH measured. Has the author ever tried measuring the other TLH in the lab to compare the values to the literature? If this is not difficult to do, it will greatly add confidence in the result. An order of magnitude increase is very significant.
- Table 2 and Figure 5 comparison. The result on Table 2 does not seem to agree with Fig. 5. At around 5 uM, the flavonoids have decreased the residual activity to 20% or less in Fig.5, but the IC50 values here and Table 2 results show that at 5 uM, the inhibition is just at 50% or less. Am I missing something?
- Line 535 to 539: Can the author comment on the fact that the protein is thermolabile, but during the purification of the protein, they undergo denaturation and renaturation. Does the author know approximately what percentage of the enzyme is active after renaturation?

Reviewer 3 ·

Basic reporting

None

Experimental design

None

Validity of the findings

Data supports the observations and conclusions drawn.

Additional comments

In this manuscript, the authors evaluated the biochemical and inhibition parameters of the recombinant VpTLH using enzymatic and hemolytic assays and determined the molecular interactions by in silico docking analysis. They concluded that flavonoids inhibit both enzymatic and hemolytic activity of thermolabile hemolysin from Vibrio Parahemolyticus. The manuscript is well written and the experiments designed well with proper controls and supports the observations and conclusions drawn with solid data. I have no major concerns with the manuscript and have few suggestions below.
1. The manuscript is difficult to read. The authors must use separate paragraphs to describe distinct experiments and ideas. This applies to the majority of the manuscript from Introduction to Discussion.
2. Figure 5, the graphs are confusing, please use different colors for the graphs.

---

## Round 0.2 · accepted · Accept

Thank you for addressing critiques of the reviewers and for amending your manuscript accordingly.